# Keeping Your Eye on the Ball:
# Trajectory Attention in Video Transformers

**Mandela Patrick**[*]
Facebook AI
mandelapatrick@fb.com

**Dylan Campbell**[*]
University of Oxford
dylan@robots.ox.ac.uk

**Yuki Asano**[*]
University of Oxford
yuki@robots.ox.ac.uk

**Ishan Misra**
Facebook AI
imisra@fb.com

**Florian Metze**
Facebook AI
fmetze@fb.com

**Christoph Feichtenhofer**
Facebook AI
feichtenhofer@fb.com

**Andrea Vedaldi**
Facebook AI
vedaldi@fb.com

**João F. Henriques**
University of Oxford
joao@robots.ox.ac.uk

## Abstract

In video transformers, the time dimension is often treated in the same way as the two spatial dimensions. However, in a scene where objects or the camera may move, a physical point imaged at one location in frame $t$ may be entirely unrelated to what is found at that location in frame $t + k$. These temporal correspondences should be modeled to facilitate learning about dynamic scenes. To this end, we propose a new drop-in block for video transformers—*trajectory attention*—that aggregates information along implicitly determined motion paths. We additionally propose a new method to address the quadratic dependence of computation and memory on the input size, which is particularly important for high resolution or long videos. While these ideas are useful in a range of settings, we apply them to the specific task of video action recognition with a transformer model and obtain state-of-the-art results on the Kinetics, Something–Something V2, and Epic-Kitchens datasets. Code and models are available at: `https://github.com/facebookresearch/Motionformer`.

## 1 Introduction

Transformers [75] have become a popular architecture across NLP [32], vision [20] and speech [5]. The self-attention mechanism in the transformer works well for different types of data and across domains. However, its generic nature and its lack of inductive biases also mean that transformers typically require extremely large amounts of data for training [56, 9], or aggressive domain-specific augmentations [71]. This is particularly true for video data, for which transformers are also applicable [50], but where statistical inefficiencies are exacerbated. While videos carry rich temporal information, they can also contain redundant spatial information from neighboring frames. Vanilla self-attention applied to videos compares pairs of image patches extracted at all possible spatial locations and frames. This can lead it to focus on the redundant spatial information rather than the temporal information, as we show by comparing normalization strategies in our experiments.

We therefore contribute a variant of self-attention, called *trajectory attention*, which is better able to characterize the temporal information contained in videos. For the analysis of still images,

---

[*]Equal contribution.

35th Conference on Neural Information Processing Systems (NeurIPS 2021).

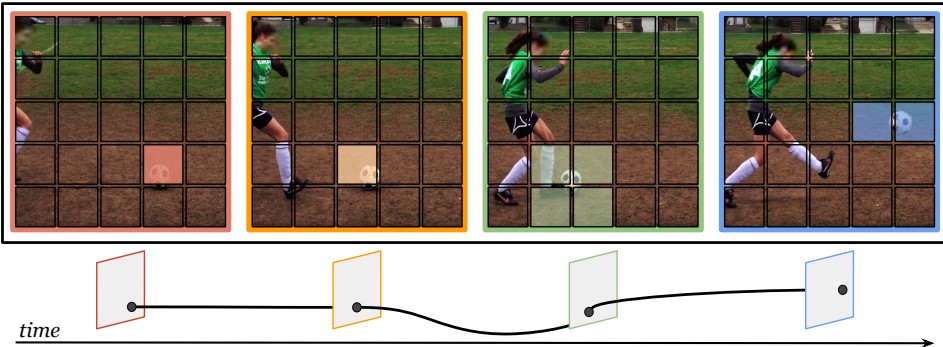

Figure 1: **Trajectory attention.** In this sequence of frames from the Kinetics-400 dataset, depicting the action 'kicking soccer ball', the ball does not remain stationary with respect to the camera, but instead moves to different locations in each frame. Trajectory attention aims to share information along the motion path of the ball, a more natural inductive bias for video data than pooling axially along the temporal dimension or over the entire space-time feature volume. This allows the network to aggregate information from multiple views of the ball, to reason about its motion characteristics, and to be less sensitive to camera motion.

spatial locality is perhaps the most important inductive bias, motivating the design of convolutional networks [42] and the use of spatial encodings in vision transformers [20]. This is a direct consequence of the local structure of the physical world: points that belong to the same 3D object tend to project to pixels that are close to each other in the image. By studying the correlation of nearby pixels, we can thus learn about the objects.

Videos are similar, except that 3D points *move* over time, thus projecting on different parts of the image along certain 2D *trajectories*. Existing video transformer methods [8, 3, 50] disregard these trajectories, pooling information over the entire 3D space-time feature volume [3, 50], or pooling axially across the temporal dimension [8]. We contend that pooling along motion trajectories would provide a more natural inductive bias for video data, allowing the network to aggregate information from multiple views of the same object or region, to reason about how the object or region is moving (for example, the linear and angular velocities), and to be invariant to camera motion.

We leverage attention itself as a mechanism to find these trajectories. This is inspired by methods such as RAFT [70], which showed that excellent estimates of optical flow can be obtained from the correlation volume obtained by comparing local features across space and time. We observe that the joint attention mechanism for video transformers computes such a correlation volume as an intermediate result. However, subsequent processing collapses the volume without consideration for its particular structure. In this work, we seek instead to use the correlation volume to guide the network to pool information along motion paths.

We also note that visual transformers operate on image patches which, differently from individual pixels, cannot be assumed to correspond to individual 3D points and thus to move along simple 1D trajectories. For example, in Figure 1, depicting the action 'kicking soccer ball', the ball spans up to four patches, depending on the specific video frame. Furthermore, these patches contain a mix of foreground (the ball) and background objects, thus at least two distinct motions. Fortunately, we are not forced to select a single putative motion: the attention mechanism allows us to assemble a motion feature from all relevant 'ball regions'.

Inspired by Nyströmformer [85], we also propose a principled approximation to self-attention, *Orthoformer*. Our approximation sets state-of-the-art performance on the recent Long Range Arena (LRA) benchmark [69] for evaluating efficient attention approximations and generalizes beyond the video domain to long text and high resolution images, with lower FLOPS and memory requirements compared to alternatives, Nyströmformer and Performer [15]. Combining our approximation with trajectory attention allows us to significantly improve its computational and memory efficiency. With our contributions, we set state-of-the-art results on four video action recognition benchmarks.

## 2   Related Work

**Video representations and 3D-CNNs.**   Hand-crafted features were originally used to convert video data into a representation amenable to analysis by a shallow linear model. Such representations include SIFT-3D [60], HOG3D [39], and IDT [76]. Since the breakthrough of AlexNet [40] on the ImageNet classification benchmark [58], which demonstrated the empirical benefits of deep neural networks to learn representations end-to-end, there have been many attempts to do the same for video. Architectures with 3D convolutions—3D-CNNs—were originally proposed to learn deep video representations [72]. Since then, improvements to this paradigm include the use of ImageNet-inflated weights [11], the space-time decomposition of 3D convolutions [54, 74, 84], channel-separated convolutions [73], non-local blocks [79], and attention layers [13]. Optical flow-based pooling can be used instead of temporal convolutions to improve the representation's robustness to camera and object motions [1]. Our approach shares this motivation.

**Vision transformers.**   The transformer architecture [75], originally proposed for natural language processing, has recently gained traction in the computer vision domain. The vision transformer (ViT) [20] decomposes an image into a sequence of $16 \times 16$ words and uses a multi-layer transformer to perform image classification. To improve ViT's data efficiency, DeiT [71] used distillation from a strong teacher model and aggressive data augmentation. Transformers have also been used in a variety of vision image tasks, such as image representation learning [12, 82, 18, 59], image generation [51], object detection [48, 10], video question-answering [35], few-shot learning [19], and image–text [49, 63, 67, 44, 68], video-text [65, 64, 89, 25, 53, 2, 6], and video-audio [43, 52, 29] representation learning.

**Attention for video recognition.**   The self-attention operation proposed in the transformer [75] have been adapted to video recognition tasks. Wang et al. [79] propose the non-local mean operation for video action recognition, which is equivalent to the standard transformer self-attention applied uniformly across space and time, while our proposed trajectory attention does not treat the space and time dimensions equivalently. Zhao et al. [87] propose a CNN architecture that explicitly predicts trajectories and aggregates information along them using a convolution operation. In contrast, our transformer architecture does not explicitly predict trajectories, but instead provides an inductive bias that encourages the network to consider motion trajectories where useful. Concurrent works [8, 3, 50, 21] have also adapted the self-attention operation to the spatio-temporal nature of videos, however, these approaches do not have a mechanism for reasoning about motion paths, treating time as just another dimension, unlike our approach.

**Efficient attention.**   Due to the quadratic complexity of self-attention, there has been a significant amount of research on how to reduce its computational complexity with respect to time and memory use. Sparse attention mechanisms [14] were used to reduce self-attention complexity to $\mathcal{O}(n\sqrt{n})$, and locality-sensitivity hashing was used by Reformer [38] to further reduce this to $\mathcal{O}(n \log n)$. More recently, linear attention mechanisms have been introduced, namely Longformer [7], Linformer [78], Performer [15] and Nyströmformer [85]. The Long Range Arena benchmark [69] was recently introduced to compare these different attention mechanisms.

**Temporal correspondences and optical flow.**   There are many approaches that aim to establish explicit correspondences between video frames as a way to reason about camera and object motion. For short-range correspondences across time, optical flow algorithms [30, 66, 70] are highly effective. In particular, RAFT [70] showed the effectiveness of an all-pairs inter-frame correlation volume as an encoding, which is essentially an attention map. All-pairs intra-frame correlations were subsequently shown to help resolve correspondence ambiguities [34]. For longer-range correspondences, object tracking by repeated detection [57] and data association can be used. In contrast to these approaches, our work does not explicitly establish temporal correspondences, but facilitates implicit correspondence learning via trajectory attention. Jabri et al. [31] estimate correspondences in a similar way, framing the problem as a contrastive random walk on a graph and apply explicit guidance via a cycle consistency loss. Incorporating such guidance into a video transformer is an interesting direction.

## 3   Trajectory Attention for Video Data

Our goal is to modify the attention mechanism in transformers to better capture the information contained in videos. Consider an input video $I \in \mathbb{R}^{T' \times 3 \times H \times W}$ consisting of $T'$ frames of resolution

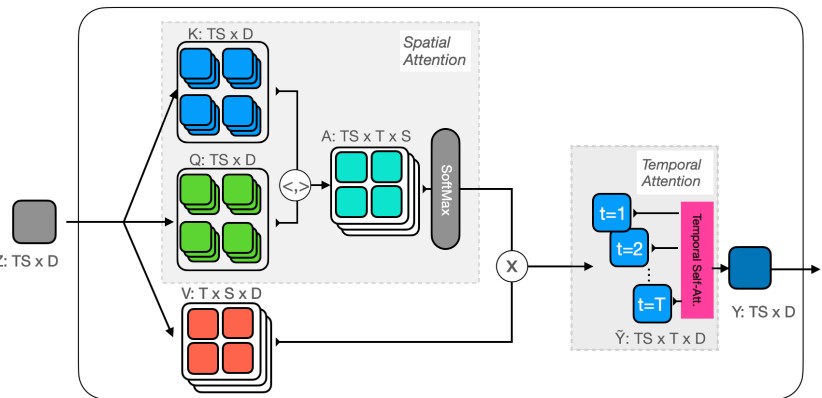

Figure 2: **Trajectory attention flowchart.** We divide the attention operation into two stages: the first forming a set of $ST$ trajectory tokens for every space-time location $st$—a spatial attention operation between pairs of frames—and the second pooling along these trajectories with a 1D temporal attention operation. In this way, we accumulate information along the motion paths of objects in the video. The softmax operations are computed over the last dimension.

$H \times W$. As in existing video transformer models [8, 3], we pre-process the video into a sequence of $ST$ tokens $\mathbf{x}_{st} \in \mathbb{R}^D$, for a spatial resolution of $S$ and a temporal resolution of $T$. We use a cuboid embedding [3, 21], where disjoint spatio-temporal cubes from the input volume are linearly projected to $\mathbb{R}^D$ (equivalent to a 3D convolution with downsampling). We also test an embedding of disjoint image patches [20]. A learnable positional encoding $\mathbf{e} \in \mathbb{R}^D$ is added to the video embeddings for spatial and temporal dimensions separately, resulting in the code $\mathbf{z}_{st} = \mathbf{x}_{st} + \mathbf{e}_s^s + \mathbf{e}_t^t$. Finally, a learnable classification token $\mathbf{z}_{\text{cls}}$ is added to the sequence of tokens, like in the BERT Transformer [32], to reason about the video as a whole. For clarity, we elide the classification token from our treatment in the sequel.

We now have a set of tokens that form the input to a sequence of transformer layers that, as in ViT [20], consist of Layer Norm (LN) operations [4], multi-head attention (MHA) [75], residual connections [27], and a feed-forward network (MLP):

$$\mathbf{y} = \text{MHA}(\text{LN}(\mathbf{z})) + \mathbf{z}; \quad \mathbf{z}' = \text{MLP}(\text{LN}(\mathbf{y})) + \mathbf{y}. \tag{1}$$

In the next section, we shall focus on a single head of the attention operation, and demonstrate how self-attention can realize a suitable inductive bias for video data. For clarity of exposition, we abuse the notation slightly, neglecting the layer norm operation and using the same dimensions for single-head attention as for multi-head attention.

## 3.1 Video self-attention

The self-attention operation begins by forming a set of query-key-value vectors $\mathbf{q}_{st}, \mathbf{k}_{st}, \mathbf{v}_{st} \in \mathbb{R}^D$, one for each space-time location $st$ in the video. These are computed as linear projections of the input $\mathbf{z}_{st}$, that is, $\mathbf{q}_{st} = \mathbf{W}_q \mathbf{z}_{st}$, $\mathbf{k}_{st} = \mathbf{W}_k \mathbf{z}_{st}$, and $\mathbf{v}_{st} = \mathbf{W}_v \mathbf{z}_{st}$, for projection matrices $\mathbf{W}_i \in \mathbb{R}^{D \times D}$. A direct application of attention across space-time (called *joint space-time attention* [8, 3]) computes:

$$\mathbf{y}_{st} = \sum_{s't'} \mathbf{v}_{s't'} \cdot \frac{\exp\langle \mathbf{q}_{st}, \mathbf{k}_{s't'} \rangle}{\sum_{\bar{s}\bar{t}} \exp\langle \mathbf{q}_{st}, \mathbf{k}_{\bar{s}\bar{t}} \rangle}. \tag{2}$$

In this way, each query $\mathbf{q}_{st}$ is compared to all keys $\mathbf{k}_{s't'}$ using dot products, the results are normalized using the softmax operator, and the weights thus obtained are used to average the values corresponding to the keys. Compared to a standard transformer, we have omitted for brevity the softmax temperature parameter $D^{1/2}$ and instead assume that the queries and keys have been divided by $D^{1/4}$.

One issue with this formulation is that it has quadratic complexity in both space and time, i.e., $\mathcal{O}(S^2 T^2)$. An alternative is to restrict attention to either space or time (called *divided space-time attention*):

$$\mathbf{y}_{st} = \sum_{s'} \mathbf{v}_{s't} \cdot \frac{\exp\langle \mathbf{q}_{st}, \mathbf{k}_{s't} \rangle}{\sum_{\bar{s}} \exp\langle \mathbf{q}_{st}, \mathbf{k}_{\bar{s}t} \rangle} \ \text{(space)}; \quad \mathbf{y}_{st} = \sum_{t'} \mathbf{v}_{st'} \cdot \frac{\exp\langle \mathbf{q}_{st}, \mathbf{k}_{st'} \rangle}{\sum_{\bar{t}} \exp\langle \mathbf{q}_{st}, \mathbf{k}_{s\bar{t}} \rangle} \ \text{(time)}. \tag{3}$$

This reduces the complexity to $\mathcal{O}(S^2T)$ and $\mathcal{O}(ST^2)$, respectively, but only allows the model to analyse time and space independently. This is usually addressed by interleaving [8] or stacking [3] the two attention modules in a sequence.

Different to both of these approaches, we perform attention *along trajectories*, the probabilistic path of a token between frames.[2] For each space-time location $st$ (the trajectory 'reference point') and corresponding query $\mathbf{q}_{st}$, we construct a set of trajectory tokens $\tilde{\mathbf{y}}_{stt'}$, representing the pooled information weighted by the trajectory probability. The trajectory extends for the duration of the video sequence and its tokens $\tilde{\mathbf{y}}_{stt'} \in \mathbb{R}^D$ at different times $t'$ are given by:

$$\tilde{\mathbf{y}}_{stt'} = \sum_{s'} \mathbf{v}_{s't'} \cdot \frac{\exp\langle \mathbf{q}_{st}, \mathbf{k}_{s't'}\rangle}{\sum_{\bar{s}} \exp\langle \mathbf{q}_{st}, \mathbf{k}_{\bar{s}t'}\rangle}. \tag{4}$$

Note that the attention in this formula is applied spatially (index $s$) and independently for each frame. Intuitively, this pooling operation implicitly seeks the location of the trajectory at time $t'$ by comparing the trajectory query $\mathbf{q}_{st}$ to the keys $\mathbf{k}_{s't'}$ at time $t'$.

Once the trajectories are computed, we further pool them across time to reason about intra-frame information/connections. To do so, the trajectory tokens are projected to a new set of queries, keys and values as usual:

$$\tilde{\mathbf{q}}_{st} = \tilde{\mathbf{W}}_q \tilde{\mathbf{y}}_{stt}, \quad \tilde{\mathbf{k}}_{stt'} = \tilde{\mathbf{W}}_k \tilde{\mathbf{y}}_{stt'}, \quad \tilde{\mathbf{v}}_{stt'} = \tilde{\mathbf{W}}_v \tilde{\mathbf{y}}_{stt'}. \tag{5}$$

Like $\mathbf{q}_{st}$ before, the updated reference query $\tilde{\mathbf{q}}_{st}$ corresponds to the trajectory reference point $st$ and contains information spatially-pooled from across the reference frame $t$. This new query is used to pool across the new time (trajectory) dimension by applying 1D attention:

$$\mathbf{y}_{st} = \sum_{t'} \tilde{\mathbf{v}}_{stt'} \cdot \frac{\exp\langle \tilde{\mathbf{q}}_{st}, \tilde{\mathbf{k}}_{stt'}\rangle}{\sum_{\bar{t}} \exp\langle \tilde{\mathbf{q}}_{st}, \tilde{\mathbf{k}}_{st\bar{t}}\rangle}. \tag{6}$$

Like joint space-time attention, our approach has quadratic complexity in both space and time, $\mathcal{O}(S^2T^2)$, so has no computational advantage and is in fact slower than divided space-time attention. However, we demonstrate better accuracy than both joint and divided space-time attention mechanisms. We also provide fast approximations in Section 3.2. A flowchart of the full trajectory attention operation is shown in tensor form in Figure 2.

## 3.2 Approximating attention

To complement our trajectory attention, we also propose an approximation scheme to speed up calculations. This scheme is generic and applies to any attention-like pooling mechanism. We thus switch to a generic transformer-like notation to describe it. Namely, consider query-key-value matrices $\mathbf{Q}, \mathbf{K}, \mathbf{V} \in \mathbb{R}^{D \times N}$ such that the query-key-value vectors are stored as columns $\mathbf{q}_i, \mathbf{k}_i, \mathbf{v}_i \in \mathbb{R}^D$ in these matrices.

In order to obtain an efficient decomposition of the attention operator, we will rewrite it using a probabilistic formulation. Let $A_{ij} \in \{0, 1\}$ be a categorical random variable indicating whether the $j$th input (with key vector $\mathbf{k}_j \in \mathbb{R}^D$) is assigned to the $i$th output (with query vector $\mathbf{q}_i \in \mathbb{R}^D$), with $\sum_j A_{ij} = \mathbf{1}$. The attention operator uses a parametric model of the probability of this event based on the multinomial logistic function, i.e., the softmax operator $\mathcal{S}(\cdot)$:[3]

$$P(A_{i:}) = \mathcal{S}(\mathbf{q}_i^\mathsf{T}\mathbf{K}), \tag{7}$$

where the subscript : denotes a full slice of the input tensor in that dimension. We now introduce the latent variables $U_{\ell j} \in \{0, 1\}$, which similarly indicate whether the $j$th input is assigned to the $\ell$th *prototype*, an auxiliary vector which we denote by $\mathbf{p}_\ell \in \mathbb{R}^D$. We can use the laws of total and conditional probability to obtain:

$$P(A_{ij}) = \sum_{\ell} P(A_{ij} \mid U_{\ell j})P(U_{\ell j}). \tag{8}$$

Note that the latent variables that we chose are independent of the inputs (keys). They use the same parametric model, but with parameters $\mathbf{P} \in \mathbb{R}^{D \times R}$ (the concatenated prototype vectors $\mathbf{p}_\ell$):

---

[2]Here, we refer to the trajectory as the motion between pairs of frames, rather than a multi-frame path.
[3]I.e. $[\mathcal{S}(\mathbf{z})]_i = \exp(z_i/\sqrt{D})/\sum_j \exp(z_j/\sqrt{D})$. For matrix inputs, the sum is over the columns.

$P(U) = \mathcal{S}(\mathbf{P}^\mathsf{T}\mathbf{K})$. Eq. 8 is *exact*, even under the parametric model for $P(U)$, though the corresponding true distribution $P(A \mid U)$ is intractable. We now *approximate* the conditional probability $P(A \mid U)$ with a similar parametric model:

$$\tilde{P}(A \mid U) = \mathcal{S}(\mathbf{Q}^\mathsf{T}\mathbf{P}), \tag{9}$$

where $\mathbf{Q} \in \mathbb{R}^{D \times N}$ concatenates all query vectors horizontally. Substituting equations 7–9 we write the full approximate attention $\tilde{\mathcal{A}}$, multiplied by an arbitrary matrix $\mathbf{V}$ (which in the case of a transformer contains the values of the key–value pairs stacked as rows):

$$\tilde{P}(A)\mathbf{V} = \mathcal{S}(\mathbf{Q}^\mathsf{T}\mathbf{P}) \left( \mathcal{S}(\mathbf{P}^\mathsf{T}\mathbf{K})\mathbf{V} \right). \tag{10}$$

**Computational efficiency.** One important feature of the approximation in eq. 10 is that it can be computed in two steps. First the values $\mathbf{V}$ are multiplied by a prototypes-keys attention matrix $\mathcal{S}(\mathbf{P}^\mathsf{T}\mathbf{K}) \in \mathbb{R}^{R \times N}$, which can be much smaller than the full attention matrix $\mathcal{S}(\mathbf{Q}^\mathsf{T}\mathbf{K}) \in \mathbb{R}^{N \times N}$ (eq. 7), i.e., $R \ll N$. Finally, this product is multiplied by a queries-prototypes attention matrix $\mathcal{S}(\mathbf{Q}^\mathsf{T}\mathbf{P}) \in \mathbb{R}^{N \times R}$, which is also small. This allows us to sidestep the quadratic dependency of full attention over the input and output size ($\mathcal{O}(N^2)$), replacing it with linear complexity ($\mathcal{O}(N)$) as long as $R$ is kept constant.

**Prototype selection.** The aim for prototype-based attention approximation schemes is to use as few prototypes as possible while reconstructing the attention operation as accurately as possible. As such, it behooves us to select prototypes efficiently. We have two priorities for the prototypes: to dynamically adjust to the query and key vectors so that their region of space is well-reconstructed, and to minimize redundancy. The latter is important because the relative probability of a query–key pair may be over-estimated if many prototypes are clustered near that query and key. To address these criteria, we incrementally build a set of prototypes from the set of queries and keys such that a new prototype is maximally orthogonal to the prototypes already selected, starting with a query or key at random. This greedy strategy is dynamic, since it selects prototypes from the current set of queries and keys, and has high entropy, since it preferences well-separated prototypes. Moreover, it balances speed and performance by using a greedy strategy, rather than finding a globally-optimal solution to the maximum entropy sampling problem [61], making it suitable for use in a transformer.

Naïvely applying prototype-based attention approximation techniques to video transformers would involve creating a unique set of prototypes for each frame in the video. However, additional memory savings can be realized by sharing prototypes across time. Since there is significant information redundancy between frames, video data is opportune for compression via temporally-shared prototypes.

**Orthoformer algorithm.** The proposed approximation algorithm is outlined in Algorithm 1. The attention matrix is approximated using intermediate prototypes, selected as the most orthogonal subset of the queries and keys, given a desired number of prototypes $R$. To avoid a linear dependence on the sequence length $N$, we first randomly subsample $cR$ queries and keys, for a constant $c$, before selecting the most orthogonal subset, resulting in a complexity quadratic in the number of prototypes $\mathcal{O}(R^2)$. The algorithm then computes two attention matrices, much smaller than the original problem, and multiplies them with the values. The most related approach in the literature is Nyströmformer [85] attention, outlined in Algorithm 2. This approach involves a pseudoinverse to attenuate the effect of near-parallel prototypes, has more operations, and a greater memory footprint.

| **Algorithm 1** Orthoformer (proposed) attention | **Algorithm 2** Nyströmformer [85] attention |
|---|---|
| 1: $\mathbf{P} \leftarrow \text{MostOrthogonalSubset}(\mathbf{Q}, \mathbf{K}, R)$ | 1: $\mathbf{P}_q, \mathbf{P}_k \leftarrow \text{SegmentMeans}(\mathbf{Q}, \mathbf{K}, R)$ |
| 2: $\mathbf{\Omega}_1 = \mathcal{S}(\mathbf{Q}^\mathsf{T}\mathbf{P}/\sqrt{D})$ | 2: $\mathbf{\Omega}_1 = \mathcal{S}(\mathbf{Q}^\mathsf{T}\mathbf{P}_k/\sqrt{D})$ |
| 3: $\mathbf{\Omega}_2 = \mathcal{S}(\mathbf{P}^\mathsf{T}\mathbf{K}/\sqrt{D})$ | 3: $\mathbf{\Omega}_2^{-1} = \text{IterativeInverse}(\mathcal{S}(\mathbf{P}_q^\mathsf{T}\mathbf{P}_k/\sqrt{D}), N_{\text{iter}})$ |
| 4: $\mathbf{Y} = \mathbf{\Omega}_1(\mathbf{\Omega}_2\mathbf{V})$ | 4: $\mathbf{\Omega}_3 = \mathcal{S}(\mathbf{P}_q^\mathsf{T}\mathbf{K}/\sqrt{D})$ |
| | 5: $\mathbf{Y} = \mathbf{\Omega}_1 \left( \mathbf{\Omega}_2^{-1} \left( \mathbf{\Omega}_3\mathbf{V} \right) \right)$ |

### 3.3 The Motionformer model

Our full video transformer model builds on previous work, as shown in Table 1. In particular, we use the ViT image transformer model [20] as the base architecture, the separate space and time positional encodings of TimeSformer [8], and the cubic image tokenization strategy as in ViViT [3]. These design choices are ablated in Section 4. The crucial difference for our model is the trajectory attention mechanism, with which we demonstrate greater empirical performance than the other models.

Table 1: **Comparison of recent video transformer models.** We show the different design choices of recent video transformer models and how they compare to our proposed Motionformer model.

| Model | Base Model | Attention | Pos. Encoding | Tokenization |
|---|---|---|---|---|
| TimeSformer [8] | ViT-B | Divided Space–Time | Separate | Square |
| ViViT [3] | ViT-L | Joint/Divided Space–Time | Joint | Cubic |
| **Motionformer** | ViT-B | Trajectory | Separate | Cubic |

## 4 Experiments

**Datasets.**  **Kinetics** [36] is a large-scale video classification dataset consisting of short clips collected from YouTube, licensed by Google under Creative Commons. As it is a dataset of human actions, it potentially contains personally identifiable information such as faces, names and license plates. **Something–Something V2** [26] is a video dataset containing more than 200,000 videos across 174 classes, with a greater emphasis on short temporal clips. In contrast to Kinetics, the background and objects remain consistent across different classes, and therefore models have to reason about fine-grained motion signals. We verified the importance of temporal reasoning on this dataset by showing that a single frame model gets significantly worse results, a decrease of 39% top-1 accuracy. In contrast, a drop of only 7% is seen on the Kinetics-400 dataset, showing that temporal information is much less relevant there. We obtained a research license for this data from `https://20bn.com`; the data was collected with consent. **Epic Kitchens-100** [16] is an egocentric video dataset capturing daily kitchen activities. The highest scoring verb and action pair predicted by the network constitutes an action, for which we report top-1 accuracy. The data is licensed under Creative Commons and was collected with consent by the Epic Kitchens teams.

**Implementation details.**  We follow a standard training and augmentation pipeline [3], as detailed in the appendix. For ablations, our default Motionformer model is the Vision Transformer Base architecture [20] (ViT/B), pretrained on ImageNet-21K [17], patch-size $2\times16\times16$ with central frame initialization [3], separate space-time positional embedding and our trajectory attention. The base architecture has 12 layers, 12 attention heads, and an embedding dimension of 768. Our default Motionformer model operates on $16\times224\times224$ videos with temporal stride 4 i.e. temporal extent of 2s. For comparisons with state-of-the-art, we report results on two additional variants: Motionformer-HR, which has a high spatial resolution ($16\times336\times336$ videos with temporal stride 4 i.e. temporal extent of 2s), and Motionformer-L, which has a long temporal range ($32\times224\times224$ videos with temporal stride 3 i.e. temporal extent of 3s). Experiments with the large ViT architecture are deferred to the appendix.

### 4.1 Ablation studies

**Input: tokenization.**  We consider the effect of different input tokenization approaches for both joint and trajectory attention on Kinetics-400 (K-400) and Something–Something V2 (SSv2) in Table 2b. For patch tokenization ($1\times16\times16$), we use inputs of size $8\times224\times224$, while for cubic [3, 21] tokenization ($2\times16\times16$), we use inputs of size $16\times224\times224$ to ensure that the model has the same number of input tokens over the same temporal range of 2 seconds. For both attention types, we see that cubic tokenization gives a 1% accuracy improvement over square tokenization on SSv2, a dataset for which temporal information is critical. Furthermore, our proposed trajectory attention using cubic tokenization outperforms joint space-time attention on both datasets.

**Input: positional encoding.**  Here, we ablate using a joint or separate [21] (default) space-time positional encoding in Table 2b. Similar to the results for input tokenization, the choice of positional encoding is particularly important for the fine-grained motion dataset, SSv2. Since joint space-time attention treats tokens in the space-time volume equally, it benefits particularly from separating the positional encodings, allowing it to differentiate between space and time dimensions, with a 4% improvement on SSv2 over joint space-time encoding. Our proposed trajectory attention elicits a more modest improvement of 1% from using separated positional encodings on SSv2, and outperforms joint space-time attention in both settings on both datasets.

**Attention block: comparisons.**  We compare our proposed trajectory attention to joint space-time attention [3], and divided space-time attention [8] in Table 4. Our trajectory attention (bottom row) outperforms both alternatives on the K-400 and SSv2 datasets. While we see only modest

Table 2: **Input encoding ablations:** Comparison of input tokenization and positional encoding design choices. We report GFLOPS and top-1 accuracy (%) on K-400 and SSv2.

(a) Cubic tokenization works best for trajectory attn.

| Attention | Tokenization | GFlops | K-400 | SSv2 |
|---|---|---|---|---|
| Joint ST | Square ($1 \times 16^2$) | 179.7 | 79.4 | 63.0 |
| | Cubic ($2 \times 16^2$) | 180.6 | 79.2 | 64.0 |
| **Trajectory** | Square ($1 \times 16^2$) | 368.5 | 79.4 | 65.8 |
| | Cubic ($2 \times 16^2$) | 369.5 | **79.7** | **66.5** |

(b) Trajectory attn. works well with both encodings.

| Attention | Pos. Encoding | GFlops | K-400 | SSv2 |
|---|---|---|---|---|
| Joint ST | Joint ST | 180.6 | 79.1 | 60.8 |
| | Separate ST [21] | 180.6 | 79.2 | 64.0 |
| **Trajectory** | Joint ST | 369.5 | 79.6 | 65.8 |
| | Separate ST [21] | 369.5 | **79.7** | **66.5** |

Table 3: **Orthoformer ablations:** We ablate various aspects of our Orthoformer approximation. E denotes exact attention and A denotes approximate attention. We report max CUDA memory consumption (GB) and top-1 accuracy (%) on K-400 and SSv2.

(a) Orthoformer is competitive with Nyström.

| Attention | Approx. | Mem. | K-400 | SSv2 |
|---|---|---|---|---|
| Trajectory (E) | N/A | 7.4 | **79.7** | **66.5** |
| Trajectory (A) | Performer | 5.1 | 72.9 | 52.7 |
| | Nyströmformer | 3.8 | **77.5** | **64.0** |
| | **Orthoformer** | 3.6 | **77.5** | 63.8 |

(b) Selecting orthogonal prototypes is the best strategy.

| Attention | Selection | Mem. | K-400 | SSv2 |
|---|---|---|---|---|
| Trajectory (E) | N/A | 7.4 | **79.7** | **66.5** |
| Trajectory (A) | Seg-Means | 3.6 | 75.8 | 60.3 |
| | Random | 3.6 | 76.5 | 62.5 |
| | **Orthogonal** | 3.6 | **77.5** | 63.8 |

(c) Approximation improves with more prototypes.

| Attention | # Prototypes | Mem. | K-400 | SSv2 |
|---|---|---|---|---|
| Trajectory (E) | N/A | 7.4 | **79.7** | **66.5** |
| Trajectory (A) | 16 | 3.1 | 73.9 | 59.2 |
| | 64 | 3.3 | 74.9 | 63.0 |
| | 128 | 3.6 | **77.5** | **63.8** |

(d) Temporal sharing is the best strategy.

| Attention | Sharing | Mem. | K-400 | SSv2 |
|---|---|---|---|---|
| Trajectory (E) | N/A | 7.4 | **79.7** | **66.5** |
| Trajectory (A) | ✗ | 16.5 | 77.3 | 61.5 |
| | ✓ | 3.6 | **77.5** | **63.8** |

improvements on the appearance cue-reliant K-400 dataset, our trajectory attention significantly outperforms ($+2\%$) the other approaches on the motion cue-reliant SSv2 dataset. This dataset requires fine-grained motion understanding, something explicitly singled out by previous video transformer works [3, 8] as a challenge for their models. In contrast, our trajectory attention excels on this dataset, indicating that its motion-based design is able to capture some of this information.

**Attention block: trajectory attention design.** We ablate two design choices for our trajectory attention: the per-frame softmax normalization and the 1D temporal attention. Unlike joint space-time attention, which normalizes the attention map over all tokens in space and time, trajectory attention normalizes independently per frame, allowing us to implicitly track the trajectories of query patches in time. In row 5 of Table 4, we ablate the benefits of this design choice. We observe a reduction of $2.5\%$ on K-400 and $5.6\%$ on SSv2 by normalizing over space and time ($\text{Norm}_{ST}$) compared with normalizing over space alone ($\text{Norm}_S$). In row 4, we show the benefit of using 1D temporal attention ($\text{Att}_T$) to aggregate temporal features, compared to average pooling ($\text{Avg}_T$). We observe reductions of $3.7\%$ on K-400 and $6.5\%$ on SSv2 when using average pooling instead of temporal attention applied to the motion trajectories, although it saves computing the additional query/key/value projections.

## 4.2 Orthoformer approximated attention

**Approximation comparisons.** In Table 3a, we compare our Orthoformer algorithm to alternative strategies: Nyströmformer [85] and Performer [15]. Our algorithm performs comparably with Nyströmformer with a reduced memory footprint. In Table 5, we also compare these attention mechanisms on the Long Range Arena benchmark [69] to show applicability to other tasks and data types. Orthoformer is able to effectively approximate self-attention, outperforming the state-of-the-art despite using far fewer prototypes (64) and so gaining significant computational and memory benefits.

**Prototype selection.** A key part of our Orthoformer algorithm is the prototype selection procedure. In Table 3b, we ablate three prototype selection strategies: segment-means, random, and greedy most-orthogonal selection. Segment-means, the strategy used in Nyströmformer, performs poorly because it can generate multiple parallel prototypes, which will over-estimate the relative probability

Table 4: **Attention ablations:** We compare trajectory attention with alternatives and ablate its design choices. We report GFLOPS and top-1 accuracy (%) on K-400 and SSv2. $\text{Att}_T$: temporal attention, $\text{Avg}_T$: temporal averaging, $\text{Norm}_{ST}$: space-time normalization, $\text{Norm}_S$: spatial normalization.

| Attention | $\text{Att}_T$ | $\text{Avg}_T$ | $\text{Norm}_S$ | $\text{Norm}_{ST}$ | GFLOPS | K-400 | SSv2 |
|---|---|---|---|---|---|---|---|
| Joint Space-Time | – | – | – | – | 180.6 | 79.2 | 64.0 |
| Divided Space-Time | – | – | – | – | 185.8 | 78.5 | 64.2 |
| | ✗ | ✓ | ✓ | ✗ | 180.6 | 76.0 | 60.0 |
| | ✓ | ✗ | ✗ | ✓ | 369.5 | 77.2 | 60.9 |
| Trajectory | ✓ | ✗ | ✓ | ✗ | 369.5 | **79.7** | **66.5** |

Table 5: **Comparison to the state-of-the-art on Long Range Arena benchmark.** GFLOPS and CUDA maximum Memory (MB) are reported for the ListOps task. Note that our algorithm achieves the best overall results with far fewer prototypes (64) than the other methods.

| Model | ListOps | Text | Retrieval | Image | Pathfinder | Avg↑ | GFLOPS↓ | Mem.↓ |
|---|---|---|---|---|---|---|---|---|
| Exact [75] | 36.69 | 63.09 | 78.22 | 31.47 | 66.35 | 55.16 | 1.21 | 4579 |
| Performer-256 [15] | 36.69 | 63.22 | **78.98** | 29.39 | **66.55** | 54.97 | 0.49 | 885 |
| Nyströmformer-128 [85] | **36.90** | 64.17 | 78.67 | **36.16** | 52.32 | 53.64 | 0.62 | 745 |
| **Orthoformer-64** | 33.87 | **64.42** | 78.36 | 33.26 | 66.41 | **55.26** | **0.24** | **344** |

of query–key pairs near those redundant prototypes. In contrast, our proposed strategy of selecting the most orthogonal prototypes from the query and key set works the best across both datasets, because it explicitly minimises prototype redundancy with respect to direction.

**Number of prototypes.** In Table 3c, we show that Orthoformer improves monotonically as the number of prototypes is increased. In particular, we see an average performance improvement of 4% on both datasets as we increase the number of prototypes from 16 to 128.

**Temporally-shared prototypes.** In Table 3d, we demonstrate the memory savings and performance benefits of sharing prototypes across time. On SSv2, we observe a 2% improvement in performance and a 5× decrease in memory usage. These gains may be attributed to the regularization effect of having prototypes leverage redundant information across frames.

**Scaling transformer models with approximated trajectory attention.** The Orthoformer attention approximation algorithm allows us to train larger models and higher resolution inputs for a given GPU memory budget. Here, we verify this, by training a large vision transformer model (ViT-L/16) [20] with a higher resolution input ($336 \times 336$ pixels) on the Kinetics-400 dataset, using the Orthoformer approximation with 196 temporally-shared prototypes and the same schedule as the base model. We use a fixed patch size (in pixels) for all models, and so the number of input tokens to the transformer scales with the square of the image resolution. As shown in Table 7, this model achieves a competitive accuracy without fine-tuning the training schedule, hyperparameters or data augmentation strategy. We expect that fine-tuning these on a validation set would greatly improve the model's performance, based on results from contemporary work [3]. Obviously such a parameter sweep is more time-consuming for these large models, however, these preliminary results are indicative that higher accuracies are attainable if these parameters were to be optimized.

### 4.3 Comparison to the state-of-the-art

In Table 6, we compare our method against the current state-of-the-art on four common benchmarking datasets: Kinetics-400, Kinetics-600, Something–Something v2 and Epic-Kitchens. We find that our method performs favorably against current methods, even when compared against much larger models such as ViViT-L. In particular, it achieves strong top-1 accuracy improvements of 1.0% and 2.3% for SSv2 and Epic-Kitchen Nouns, respectively. These datasets require greater motion reasoning than Kinetics and so are a more challenging benchmark for video action recognition.

## 5 Conclusion

We have presented a new general-purpose attention block for video data that aggregates information along implicitly determined motion trajectories, lending a realistic inductive bias to the model. We further address its quadratic dependence on the input size with a new attention approximation

Table 6: **Comparison to the state-of-the-art on video action recognition.** We report GFLOPS and top-1 (%) and top-5 (%) video action recognition accuracy on K-400/600, and SSv2. On Epic-Kitchens, we report top-1 (%) action (A), verb (V), and noun (N) accuracy.

(a) **Something–Something V2**

| Model | Pretrain | Top-1 | Top-5 | GFLOPs ×views |
|---|---|---|---|---|
| SlowFast [24] | K-400 | 61.7 | - | 65.7×3×1 |
| TSM [47] | K-400 | 63.4 | 88.5 | 62.4×3×2 |
| STM [33] | IN-1K | 64.2 | 89.8 | 66.5×3×10 |
| MSNet [41] | IN-1K | 64.7 | 89.4 | 67×1×1 |
| TEA [46] | IN-1K | 65.1 | - | 70×3×10 |
| bLVNet [22] | IN-1K | 65.2 | 90.3 | 128.6×3×10 |
| VidTr-L [45] | IN-21K+K-400 | 60.2 | - | 351×3×10 |
| Tformer-L [8] | IN-21K | 62.5 | - | 1703×3×1 |
| ViViT-L [3] | IN-21K+K-400 | 65.4 | 89.8 | 3992×4×3 |
| MViT-B [21] | K-400 | 67.1 | 90.8 | 170×3×1 |
| **Mformer** | IN-21K+K-400 | 66.5 | 90.1 | 369.5×3×1 |
| **Mformer-L** | IN-21K+K-400 | **68.1** | **91.2** | 1185.1×3×1 |
| **Mformer-HR** | IN-21K+K-400 | 67.1 | 90.6 | 958.8×3×1 |

(b) **Kinetics-400**

| Method | Pretrain | Top-1 | Top-5 | GFLOPs ×views |
|---|---|---|---|---|
| I3D [11] | IN-1K | 72.1 | 89.3 | 108×N/A |
| R(2+1)D [74] | - | 72.0 | 90.0 | 152×5×23 |
| S3D-G [84] | IN-1K | 74.7 | 93.4 | 142.8×N/A |
| X3D-XL [23] | - | 79.1 | 93.9 | 48.4×3×10 |
| SlowFast [24] | - | 79.8 | 93.9 | 234×3×10 |
| VTN [50] | IN-21K | 78.6 | 93.7 | 4218×1×1 |
| VidTr-L [45] | IN-21K | 79.1 | 93.9 | 392×3×10 |
| Tformer-L[8] | IN-21K | 80.7 | 94.7 | 2380×3×1 |
| MViT-B [21] | - | 81.2 | 95.1 | 455×3×3 |
| ViViT-L [3] | IN-21K | 81.3 | 94.7 | 3992×3×4 |
| **Mformer** | IN-21K | 79.7 | 94.2 | 369.5×3×10 |
| **Mformer-L** | IN-21K | 80.2 | 94.8 | 1185.1×3×10 |
| **Mformer-HR** | IN-21K | 81.1 | **95.2** | 958.8×3×10 |

(c) **Epic-Kitchens**

| Method | Pretrain | A | V | N |
|---|---|---|---|---|
| TSN [77] | IN-1K | 33.2 | 60.2 | 46.0 |
| TRN [88] | IN-1K | 35.3 | 65.9 | 45.4 |
| TBN [37] | IN-1K | 36.7 | 66.0 | 47.2 |
| TSM [47] | IN-1K | 38.3 | **67.9** | 49.0 |
| SlowFast [24] | K-400 | 38.5 | 65.6 | 50.0 |
| ViViT-L [3] | IN-21K+K-400 | 44.0 | 66.4 | 56.8 |
| **Mformer** | IN-21K+K-400 | 43.1 | 66.7 | 56.5 |
| **Mformer-L** | IN-21K+K-400 | 44.1 | 67.1 | 57.6 |
| **Mformer-HR** | IN-21K+K-400 | **44.5** | 67.0 | **58.5** |

(d) **Kinetics-600**

| Model | Pretrain | Top-1 | Top-5 | GFLOPs ×views |
|---|---|---|---|---|
| AttnNAS [80] | - | 79.8 | 94.4 | - |
| LGD-3D [55] | IN-1K | 81.5 | 95.6 | - |
| SlowFast [24] | - | 81.8 | 95.1 | 234×3×10 |
| X3D-XL [23] | - | 81.9 | 95.5 | 48.4×3×10 |
| Tformer-HR [8] | IN-21K | 82.4 | 96.0 | 1703×3×1 |
| ViViT-L [3] | IN-21K | 83.0 | 95.7 | 3992×3×4 |
| MViT-B-24 [21] | - | **83.8** | **96.3** | 236×1×5 |
| **Mformer** | IN-21K | 81.6 | 95.6 | 369.5×3×10 |
| **Mformer-L** | IN-21K | 82.2 | 96.0 | 1185.1×3×10 |
| **Mformer-HR** | IN-21K | 82.7 | 96.1 | 958.8×3×10 |

Table 7: **Can we train larger models using approximated trajectory attention?** We report top-1 and top-5 accuracy (%) on the Kinetics-400 dataset of two variants of our Motionformer model: Motionformer-B and Motionformer-H. The former uses the base model with exact (E) trajectory attention, while the latter uses a much larger model (ViT-L) and a higher resolution input (336 × 336 pixels) with approximate (A) trajectory attention, i.e., using Orthoformer. We reduce this to a linear relationship with the Orthoformer approximation, which allows us to fit the model on the GPU.

| Model | Base model | Params | Attention | Max memory (GB) | Top-1 | Top-5 |
|---|---|---|---|---|---|---|
| Mformer-B | ViT-B/224 | 109.1M | Trajectory (E) | **7.3** | 79.7 | 94.2 |
| Mformer-H | ViT-L/336 | 381.9M | Trajectory (A) | 22.2 | **80.0** | **94.5** |

algorithm that significantly reduces the memory requirements, the largest bottleneck for transformer models. With these contributions, we obtain state-of-the-art results on several benchmark datasets. Nonetheless, our approach inherits many of the limitations of transformer models, including poor data efficiency and slow training. Specific to this work, trajectory attention has higher computational complexity than alternative attention operations used for video data. This is attenuated by the proposed approximation algorithm, with significantly reduced memory and computation requirements. However, its runtime is bottlenecked by prototype selection, which is not easily parallelized.

**Future work.** There are many applications of trajectory attention beyond video action classification, such as those tasks where temporal context is highly important. We see significant potential for using trajectory attention for tracking [28], temporal action localization [83, 81] and online action detection [86, 62], among other settings, and leave these as avenues for future work.

**Potential negative societal impacts.** One negative impact of this research is the significant environmental impact associated with training transformers, which are large and compute-expensive models. Compared to 3D-CNNs where the compute scales linearly with the sequence length, video transformers scale quadratically. To mitigate this, we proposed an approximation algorithm with linear complexity that greatly reduces the computational requirements. There is also potential for video action recognition models to be misused, such as for unauthorized surveillance.

## Acknowledgments and Disclosure of Funding

We are grateful for support from the Rhodes Trust (M.P.), the European Research Council Starting Grant (IDIU 638009, D.C.), Qualcomm Innovation Fellowship (Y.A.), the Royal Academy of Engineering (RF201819/18/163, J.H.), and EPSRC Centre for Doctoral Training in Autonomous Intelligent Machines & Systems (EP/L015897/1, M.P. and Y.A.). Funding for M.P. was received under his Oxford affiliation. We thank Bernie Huang, Dong Guo, Rose Kanjirathinkal, Gedas Bertasius, Mike Zheng Shou, Mathilde Caron, Hugo Touvron, Benjamin Lefaudeux, Haoqi Fan, and Geoffrey Zweig from Facebook AI for their help, support, and discussion around this project. We also thank Max Bain and Tengda Han from VGG for fruitful discussions.

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
