# Keeping Your Eye on the Ball:
# Trajectory Attention in Video Transformers
# Appendix

**Mandela Patrick**[*]
Facebook AI
mandelapatrick@fb.com

**Dylan Campbell**[*]
University of Oxford
dylan@robots.ox.ac.uk

**Yuki Asano**[*]
University of Oxford
yuki@robots.ox.ac.uk

**Ishan Misra**
Facebook AI
imisra@fb.com

**Florian Metze**
Facebook AI
fmetze@fb.com

**Christoph Feichtenhofer**
Facebook AI
feichtenhofer@fb.com

**Andrea Vedaldi**
Facebook AI
vedaldi@fb.com

**João F. Henriques**
University of Oxford
joao@robots.ox.ac.uk

## .1 Further experimental analysis and results

### .1.1 Does trajectory attention make better use of motion cues?

In the main paper (and below in Section .1.2), we provide evidence that action classification on the Something–Something V2 (SSv2) dataset [8] is more reliant on motion cues than the Kinetics dataset [10], where appearance cues dominate and a single-frame model achieves high accuracy. Improved performance on SSv2 is one way to infer that our model makes better use of temporal information, however, here we consider another way. We artificially adjust the speed of the video clips by changing the temporal stride of the input. A larger stride simulates faster motions, with adjacent frames being more different. If our trajectory attention is able to make better use of the temporal information in the video than the other attention mechanisms, we expect the margin of improvement to increase as the temporal stride increases. As shown in Figure 1, this is indeed what we observe, with the lines diverging as temporal stride increases, especially for the motion cue-reliant SSv2 dataset. Since the same number of frames are used as input in all cases, the larger the stride, the more of the video clip is seen by the model. This provides additional confirmation that seeing a small part of a Kinetics video is usually enough to classify it accurately, as shown on the bottom left, where the absolute accuracy is reported.

### .1.2 How important are motion cues for classifying videos from the Kinetics-400 and Something–Something V2 datasets?

To determine the relative importance of motion cues compared to appearance cues for classifying videos on two of the major video action recognition datasets (Kinetics-400 and Something–Something V2), we trained a single frame vision transformer model and compare the results to a multi-frame model that can reason about motion. The single frame was sampled from the video at random. Table 1 shows that single-frame action classifiers can do almost as well as video action classifiers on the Kinetics-400 dataset, implying that the motion information is much less relevant. In contrast, classifying videos from the Something-Something V2 dataset clearly requires this motion information. Therefore, to excel on the SSv2 dataset, a model must reason about motion information. Our model,

---

[*] Equal contribution.

35th Conference on Neural Information Processing Systems (NeurIPS 2021), Sydney, Australia.

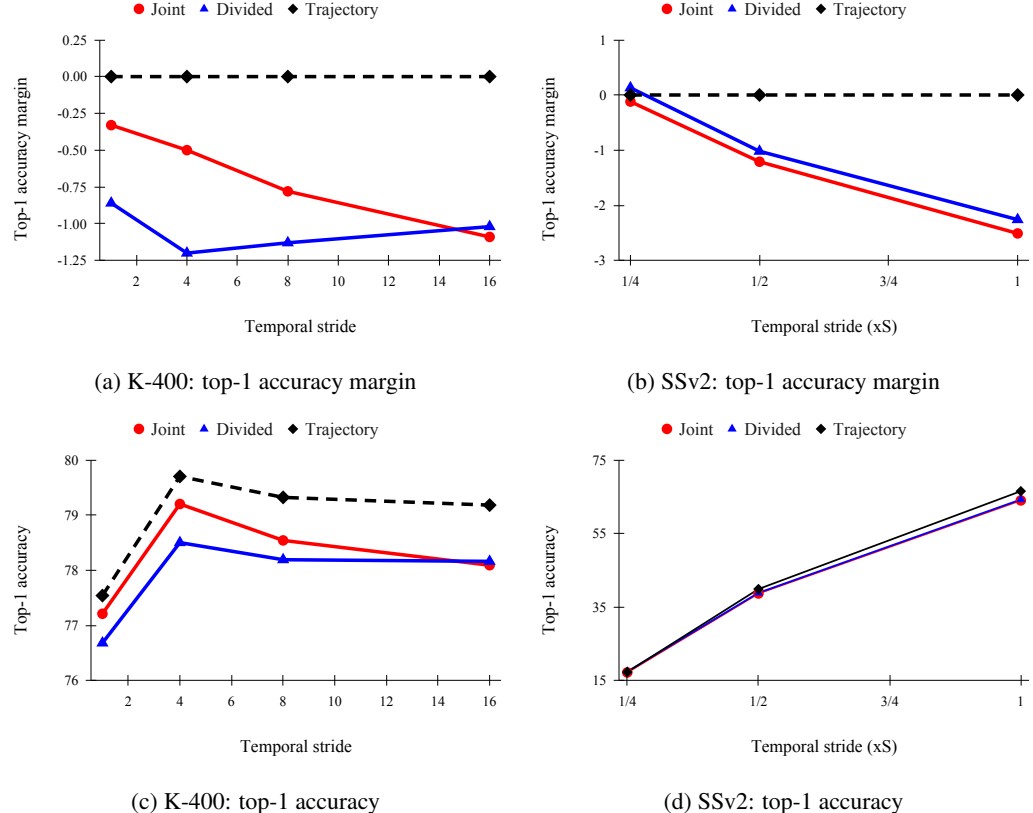

(a) K-400: top-1 accuracy margin

(b) SSv2: top-1 accuracy margin

(c) K-400: top-1 accuracy

(d) SSv2: top-1 accuracy

Figure 1: **Does trajectory attention make better use of motion cues?** Performance of transformer models with joint space-time attention, divided space-time attention, and trajectory attention, as the temporal stride increases, on the Kinetics-400 dataset (left) and the Something–Something V2 dataset (right). Top: top-1 accuracy margin relative to trajectory attention (difference of accuracy and trajectory accuracy). Bottom: absolute top-1 accuracy shown for reference. If our trajectory attention is able to make better use of the temporal information in the video than the other attention mechanisms, we expect the accuracy margin between the methods to increase as the temporal stride increases. This is indeed the observed behaviour, especially for the motion cue-reliant SSv2 dataset. A larger stride simulates greater motion between input frames, which trajectory attention is better able to model and reason about. Note that the larger the stride, the more of the video clip is seen by the model; for all plots, the rightmost side of the axis corresponds to the entire video clip. Note also that the strides for SSv2 are written as multiples of $S$, the stride needed to evenly sample the entire video clip.

which introduces an inductive bias that favors pooling along motion trajectories, is able to do this and sees corresponding performance gains.

### .1.3 Which classes is the performance difference larger with and without the trajectory attention?

The class labels with the largest performance increase (given in parentheses) on the Something-Something v2 dataset are: "Spilling [something] next to [something]" (18%), "Pretending to put [something] underneath [something]" (15%), and "Trying to pour [something] into [something], but missing so it spills next to it" (14%). The classes with the largest performance decrease are: "Putting [something] that can't roll onto a slanted surface, so it stays where it is" (10%), "Putting [something] on a flat surface without letting it roll" (9%), and "Showing a photo of [something] to the camera" (8%). It is apparent that classes involving predominantly stationary objects do not benefit from trajectory attention, as we would expect.

Table 1: **Importance of motion cues for the K-400 and SSv2 datasets.** A classifier for the K-400 dataset performs well when all motion information is removed (1 frame model), while a classifier for the SSv2 dataset performs very poorly. Therefore, SSv2 is a better dataset for evaluating *video* action classification, where the combination of appearance and motion is critical.

| Dataset | Top-1 accuracy (1 frame) | Top-1 accuracy (8 frames) | $\Delta$ |
|---|---|---|---|
| Kinetics-400 | 73.2 | 79.7 | 6.5 |
| Something–Something V2 | 27.1 | 66.5 | **39.4** |

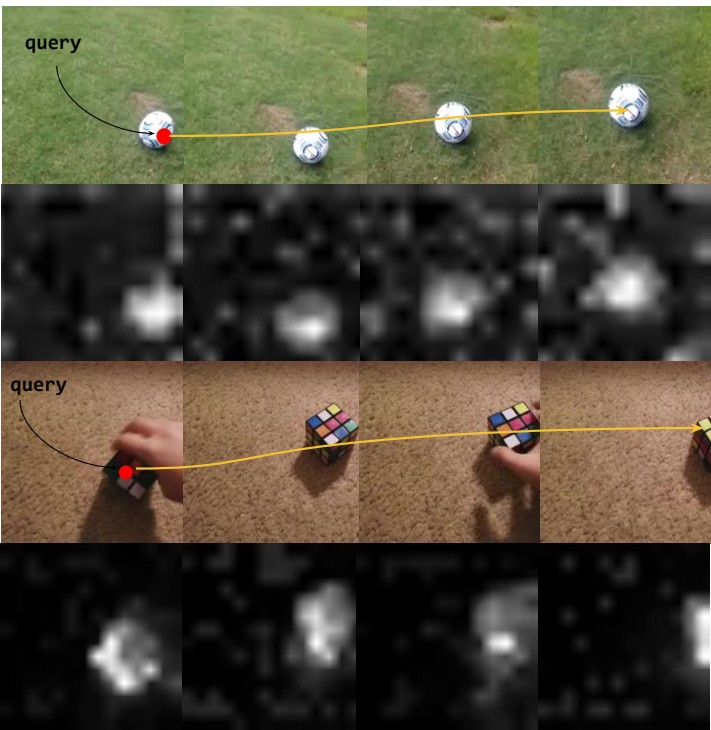

Figure 2: **Trajectory attention maps.** In this sequence of frames from Kinetics-400 (row 1) and Something-Something V2 (row 3), we show the attention maps at each frame given an initial query point (red point). We see that the model learns to implicitly track along motion paths (yellow arrow) using our trajectory attention module.

### .1.4 Trajectory attention maps

In Figure 2, we show qualitative results of the intermediate attention maps of our trajectory attention operation. The learned attention maps appear to implicitly track the query points across time, a strategy that is easier to learn with the inductive bias instilled by trajectory attention.

### .1.5 How long does it take to train Motionformer model?

For Table 3c, using Motionformer with orthoformer approximation, 16 prototypes take 384 GPU hours, 64 prototypes take 800 GPU hours, and 128 prototypes take 1216 GPU hours. For the Kinetics-400 state-of-the-art table, the Mformer-B model took 384 GPU hours, the Mformer-L took 1334 GPU hours, and Mformer-HR model took 1376 GPU hours to train. Our baseline Mformer-B model, which outperforms TimeSformer-B by over $1\%$, takes similar GPU hours (416 (ours) vs the 416 GPU hours reported in Table 2 of TimeSformer). We cannot directly compare to ViViT because they didn't

report training time, but they used a very large transformer (24 layers) compared to ours (12 layers) and so we expect the training time for their approach to be significantly greater.

## .1.6 Semi-supervised Video Object Segmentation on DAVIS 2017

We evaluate our baseline Motionformer Kinetics-pretrained model (16x16 with Trajectory Attention) on the semi-supervised video object segmentation task on the DAVIS 2017 dataset as in Jabri et al. [9] in Table 2. We directly use the attention maps of our Motionformer model in the label propagation setting, as in [9]. We report mean (m) of standard boundary alignment (F) and region similarity (J) metrics. We attain a competitive J&F-Mean of 60.6. For comparison, DINO [3] obtains J&F-Mean of 62.3 with the same architecture (ViT-B/16x16), but by using a self-supervised learning task on IM-1K. We expect that we could significantly improve the performance by using an 8x8 patch size, as this was shown to be highly effective for the task [3].

Table 2: **DAVIS 2017 Video object segmentation.** We evaluate the quality of frozen features on video instance tracking. We report mean region similarity $\mathcal{J}_m$ and mean contour-based accuracy $\mathcal{F}_m$.

| Method | Data | Arch. | $(\mathcal{J}\&\mathcal{F})_m$ | $\mathcal{J}_m$ | $\mathcal{F}_m$ |
|---|---|---|---|---|---|
| *Supervised* | | | | | |
| ImageNet | INet | ViT-S/8 | 66.0 | 63.9 | 68.1 |
| STM [14] | I/D/Y | RN50 | 81.8 | 79.2 | 84.3 |
| Ours | K-400 | Mformer-B/16 | 60.6 | 58.3 | 62.9 |
| *Self-supervised* | | | | | |
| CT [17] | VLOG | RN50 | 48.7 | 46.4 | 50.0 |
| MAST [11] | YT-VOS | RN18 | 65.5 | 63.3 | 67.6 |
| STC [9] | Kinetics | RN18 | **67.6** | **64.8** | **70.2** |
| DINO [3] | INet | ViT-B/16 | 62.3 | 60.7 | 63.9 |

## .2 Implementation details

**Preprocessing.** During training, we randomly sample clips of size $16 \times 224 \times 224$ at a rate of $1/4$ from 30 FPS videos, thereby giving an effective temporal resolution of just over 2 seconds. We normalize the inputs with mean and standard deviation 0.5, rescaling in the range $[-1, 1]$. We use standard video augmentations such as random scale jittering, random horizontal flips and color jittering. For smaller datasets such as Something–Something V2 and Epic-Kitchens, we additionally apply rand-augment [5]. During testing, we uniformly sample 10 clips per video and apply a 3 crop evaluation [7].

**Training.** For all datasets, we use the AdamW [12] optimizer with weight decay $5 \times 10^{-2}$, a batch size per GPU of 4, label smoothing [15] with alpha 0.2 and mixed precision training [13]. For Kinetics-400/600 and Something-Something V2, we train for 35 epochs, with an initial learning rate of $10^{-4}$, which we decay by 10 at epochs 20, 30. As Epic-Kitchens is a smaller dataset, we use a longer schedule and train for 50 epochs with decay at 30 and 40.

**Long Range Arena benchmark details.** For the Long-Range Arena benchmark [16], we used the training, validation, and testing code and parameters from the Nyströmformer Github repository. The Performer [4] implementation was ported over to PyTorch from the official Github repo, and the Nyströmformer [19] implementation was used directly from its Github repository.

**Computing resources.** Ablation experiments were run on a GPU cluster using 4 nodes (32 GPUs) with an average training time of 12 hours. Experiments for comparing with state-of-the-art models used 8 nodes (64 GPUs), with an average training time of 7 hours.

**Libraries.** For our code implementation, we used the `timm` [18] library for our base vision transformer implementation, and the `PySlowFast` [6] library for training, data processing, and the evaluation pipeline.