# OpenReview forum: "Keeping Your Eye on the Ball: Trajectory Attention in Video Transformers"
_NeurIPS.cc/2021/Conference — NeurIPS 2021 Oral_

### Official Review · Reviewer_E1g8 · 2021-07-15

**Rating:** 7
**Confidence:** 3

**Summary:**

The paper introduces a novel transformer architecture in which trajectories in video are considered. The model is able to leverage its own self-attention module to build trajectories around relevant objects in the video. Furthermore, authors introduce an approximation through a set of intermediate variables which enables them to reduce the computational cost of the transformer. Authors validate their model against state-of-the-art models in three main video benchmarks.

**Limitations And Societal Impact:**

The authors discuss the societal impact and limitations of the work in the paper.

**Main Review:**

Summary: The architecture proposed is interesting and well validated through ablation experiments. Furthermore, authors show that the proposed model improves performance over the baselines. Finally, the approximation technique introduced is useful to reduce the computational needs. For all those reasons, I believe the paper should be accepted.

Strengths:

- The paper is well written and the novelty aspects are well highlighted and contextualised.

- Introducing elements in the transformer which relates to the nature of the input is an interesting and useful exercise. The community has commonly relied on the transformer architecture as a black box and authors are introducing some previous knowledge in order to improve the performance by tracking trajectories through time. I think this is a good direction and authors show that can have positive impact to the final outcome of the model.

- The paper presents a good ablation study to better understand the design decisions through the paper. Furthermore, they report GFLOPS and memory used, which is very useful for the reader to understand the different models and its advantages and disadvantages.

- The approximation through prototype vectors is a simple but effective mesure to reduce computational complexity. I think this idea can be useful for the community to not only improve this model but also other with similar operations on it.

- The results compared to the state-of-the-art models are strong, specially in the Something-Something and EpicKitchens benchmarks.

Weaknesses / Things to improve:

- The paper story relies on the idea that the model is able to track trajectories of objects through the video. However, only a figure in the supplemental show examples of this tracked objects. I wonder if authors could quantitatively evaluate the tracking ability with a tracking dataset.

- Some of the ablations such as Input Tokenisation or Position Encoding are not strongly related with the main innovation of the paper. I believe that it's very interesting to report those numbers for the readers, but moving those experiments to supplemental would give authors a bit more space to discuss some of the results in the supplemental such as scaling-up the transformer models like in Table 2 of the supplemental.

- It would be interesting for the reader to see a more detailed analysis on the classes where the performance difference is larger with and without the trajectory block. I think it would improve the understanding of the model by the reader.

**Time Spent Reviewing:**

3

---

> ### Author Response · Authors · 2021-08-09
> **Response to Reviewer E1g8**
>
> Thank you for your review. We are pleased that you find the proposed architecture interesting, well-validated, and performant, and the approximation scheme simple but effective at reducing the computational complexity. We will consider each of your concerns below.
>
> **Tracking results**: Thank you for the suggestion. We have evaluated our baseline Motionformer Kinetics-pretrained model (16x16 with Trajectory Attention) on the semi-supervised video object segmentation task on the DAVIS 2017 dataset as in Jabri et al. [1]. We directly use the attention maps of our Motionformer model in the label propagation setting, as in [1]. We report mean (m) of standard boundary alignment (F) and region similarity (J) metrics. We attain a competitive J&F-Mean of 60.6. For comparison, DINO [2] obtains J&F-Mean of 62.3 with the same architecture (ViT-B/16x16), but by using a self-supervised learning task on IM-1K. We expect that we could significantly improve the performance by using an 8x8 patch size, as this was shown to be highly effective for the task [2]. We will add these results to the revised paper.
> [1]: Space-Time Correspondence as a Contrastive Random Walk. Jabri et al. NeurIPS 2020.
> [2]: Emerging Properties in Self-Supervised Vision Transformers. Caron et al. ArXiv 2021.
>
> **Large model results in main paper**: Thank you for the recommendation. Yes, we will move the discussion on scaling up transformer models into the main paper for the final version.
>
> **Class-specific analysis**: Thank you for the recommendation. We will include this class-specific analysis, comparing the models with and without trajectory attention. The class labels with the largest performance increase (given in parentheses) on the Something-Something v2 dataset are: “Spilling [something] next to [something]” (18%), “Pretending to put [something] underneath [something]” (15%), and “Trying to pour [something] into [something], but missing so it spills next to it” (14%). The classes with the largest performance decrease are: “Putting [something] that can't roll onto a slanted surface, so it stays where it is” (10%), “Putting [something] on a flat surface without letting it roll” (9%), and “Showing a photo of [something] to the camera” (8%). It is apparent that classes involving predominantly stationary objects do not benefit from trajectory attention, as we would expect.

---

> > ### Comment · Reviewer_E1g8 · 2021-08-23
> > **Thanks!**
> >
> > Hi,
> >
> > First, I would like to thank the authors for the detailed response and the additional experiments they provide in their rebuttal. After reading the other reviews and rebuttals, I think this is a strong paper and it should be accepted.
> >
> > Best wishes,

---

### Official Review · Reviewer_8Cfh · 2021-07-15

**Rating:** 7
**Confidence:** 4

**Summary:**

The paper focused on model related motion information along the temporal domain. In particular, they proposed new drop-in block for video transformers that aggregates information along implicitly determined motion paths. The network is based on attentive mechanisms with Transformer. The new model doesn't rely on quadratic dependence of computation and memory on the input size, and is efficient to be applied to high resolution or long videos. The model was evaluated on multiple standard benchmarks, and it achieved convincing results.

**Ethical Concerns:**

I didn't see any concern about this.

**Ethics Review Area:**

["I don’t know"]

**Limitations And Societal Impact:**

Yes, limitations are explained.

**Main Review:**

Strengths:

- The idea of modeling temporal information along the object moving trajectory is convincing.

- The paper writing is fairly good, and the proposed method is novel.

- The proposed method was evaluated on most of standard action recognition benchmarks, and the performance is reasonable and convincing.

Weakness:

- The qualitative results provided in the appendix is too simple. There is only one moving object in the videos, and the attention mainly focuses on the object. It's better to show some complicated cases that at least multiple objects are moving and their corresponding attention maps.

- The proposed method is only evaluated on action recognition, which is not a problem. But as the authors mentioned that their method is efficient and effective for long videos, in the future, it is interesting to show its performance on some untrimmed video tasks, such as temporal action localization [1,2] and online action detection [3,4]. As they are more challenging due to the complicated temporal context, and modeling with trajectory attention could benefit more on these tasks.

[1] Long-Term Feature Banks for Detailed Video Understanding. Wu et al. CVPR.

[2] Learning to track for spatio-temporal action localization. Weinzaepfel et al. ICCV.

[3] Long Short-Term Transformer for Online Action Detection. Xu et al. arXiv.

[4] Online Real-time Multiple Spatiotemporal Action Localisation and Prediction. Singh et al.

- Trajectory convolution [5] shares similar ideas with the proposed method. Reviewing it as related work would be necessary.

[5] Trajectory Convolution for Action Recognition. Zhao et al. NeurIPS.

- It seems that the attention focuses on the "key" moving object in the video, could the authors explain more about the difference between the proposed method and non-local?

**Time Spent Reviewing:**

5 hours

---

> ### Author Response · Authors · 2021-08-09
> **Response to Reviewer 8Cfh**
>
> Thank you for your review. We are encouraged that you find the idea of modeling temporal information along trajectories convincing and novel, and the performance properly evaluated. We will consider each of your concerns below.
>
> **Attention maps for multiple moving objects**: We presented a simple example in the appendix to illustrate the intuition behind trajectory attention in the clearest way, but will add additional, more complex examples in the revision. In general, the attention distribution for objects with a distinctive appearance is more peaked than for less distinctive objects, and (all other things being equal) larger objects have a stronger response than small objects, relative to the patch size.
>
> **Applications of trajectory attention**: Yes, we agree that there are many applications of trajectory attention beyond video action classification, such as those tasks suggested by the reviewer where temporal context is highly important. We see significant potential for using trajectory attention for tracking, trajectory prediction, temporal action localization and online action detection, among other settings. Thank you for the suggestions - we will add these as avenues for future work in the conclusion.
>
> **Additional citation**: Agreed, [5] is quite related - we will add the following to the related works section: “Zhao et al. [5] propose a CNN architecture that explicitly predicts trajectories and aggregates information along them using a convolution operation. In contrast, our transformer architecture does not explicitly predict trajectories, but instead provides an inductive bias that encourages the network to consider motion trajectories where useful. This has several advantages, allowing a simpler architecture, probabilistic trajectories, and non-coherent trajectories (such as where multiple objects are co-located in an input patch of one frame and move independently).”
> If an explicit approach, such as [5], were used, we would need a much higher resolution of input to the transformer, with concomitant computation increases, and a much more accurate motion field prediction network, such as an optical flow network.
>
> **Comparison with non-local neural networks**: The non-local mean operations considered in Wang et al. [2018] are equivalent to standard transformer self-attention (as noted in their paper: “the self-attention module [49] recently presented for machine translation is a special case of non-local operations in the embedded Gaussian version”) and three other variations (equivalent to self-attention without projection, self-attention without softmax, and a learned affinity function). As such, the method is equivalent to joint space-time attention, when considering video data. In contrast, trajectory attention does not treat space and time dimensions equivalently. Instead, it first constructs trajectory tokens using an inter-frame attention operation between every pair of frames (unlike self-attention, the queries and keys are from different frames) and then pools the trajectory tokens in a second 1D attention operation. This encourages information to flow along motion trajectories, which is not the case for non-local neural networks. We also note that neither our work nor non-local neural networks focus on a single key moving object, although we both use such an example for illustrative purposes. We will add this comparison to the revised paper.

---

> > ### Comment · Reviewer_8Cfh · 2021-08-23
> > **Final Comments**
> >
> > The authors adequately solved my main concerns in the rebuttal. After checking other reviewers' comments and the rebuttal, I will keep my rating as Accept.

---

### Official Review · Reviewer_CQwE · 2021-07-19

**Rating:** 7
**Confidence:** 3

**Summary:**

This work proposed a novel drop-in block for video transformers. Different from transformers that work on image or language modality, this block aggregates information along motion paths. The key idea is to design an efficient algorithm to approximation the path (trajectory attention), which requires quadratic dependence of computation and memory on the input size. As a solution, the proposed Motionformer model uses the (1) ViT image transformer model as the base architecture, (2) the separate space and time positional encodings of TimeSformer, (3) and the cubic image tokenization strategy as in ViViT. Experimental results on four video action recognition benchmarks show Motionformer can achieve state-of-the-art performance.

**Limitations And Societal Impact:**

As explained in Section 5, Motionformer has its limitation of poor data efficiency and slow training, which is common for transformer-based methods. Since this work needs to compute trajectory attention, it has higher computational complexity.

**Main Review:**

(+) This work brings a novel idea to video understanding. It implicitly determined motion paths by trajectory attention. It allows the network to (1) aggregate information from multiple views of the object/region, (2) reason the movement of the object/region, and (3) be invariant to camera motion.

(+) The approximation scheme can speed up the calculation of trajectory attention. It reduces the complexity in time and space, which is essential in transformer-based models.

(+) Extensive experiments are conducted on popular video action recognition benchmarks, justifying the effectiveness of this work.

(+) The authors also provide code in the supplementary material. It is beneficial for the video understanding community.

**Time Spent Reviewing:**

4 hours

---

> ### Author Response · Authors · 2021-08-09
> **Response to Reviewer CQwE**
>
> Thank you for your review and the encouraging comments about our work bringing “a novel idea [trajectory attention] to video understanding”. We agree with your assessment of the limitations inherited from the transformer architecture (poor data efficiency and slow training), and the higher computational complexity of our method. We appreciate the time you have spent reading and reviewing our paper.

---

### Official Review · Reviewer_m5y4 · 2021-07-20

**Rating:** 7
**Confidence:** 2

**Summary:**

The paper proposes a new component for transformer architectures to model trajectory attention by modeling temporal correspondences in dyanmic scenes. They also present approximation techniques to address computation and memory challenges for high-resolution videos. The proposed contributions are demonstrated on video action recognition tasks with SOTA results on multiple datasets.

**Limitations And Societal Impact:**

I would encourage the authors to provide more commentary on how the trajectory attention model and approximation techiques can be applied to other domains and tasks which benefit from modeling temporal correspondence.


**Main Review:**

Main contributions:
- a new block which can be easily integrated into transformer architectures to mode the temporal correpsondence in dynamic scenes (trajectory attention)
- approximation techqniques (Orthoformer algorithm) to reduce memory usage and minimize quadratic dependency to facilate the handling of long duration, high resolution videos
- Integration of these contributions into Motionformer model and demonstration on video action recognition task to achieve SOTA results on Kinetics, Something-Something V2, and Epic-Kitchens datasets.

The trajectory attention is implemented by using the two stages (a spatial attention operation and a 1D temporal attention) as shown in the Figure2.  For the trajectory attention test, the paper proposed Motionformer model which is using the VIT image transformer model as the base architecture and the trajectory attention for the attention mechanism. Through ablation studies, the paper proved that the trajectory attention showed better accuracy than other attention methods (Table 4). In particular, there was a significant improvement in the SSv2 data test. This shows that the proposed model with the trajectory attention can understand fine-grained motion signals more effectively than other attention mechanisms.
The limitation of the trajectory attention occurred when it computed the trajectory attention because it’s causing quadratic complexity in both space and time. To overcome this, the approximation algorithm is proposed. It was shown that the proposed approximation algorithm used less memory than other models and showed better results overall in Tables 3 and 5.
At the end, Table 6 showed the proposed model (Motionformer) achieved comparable or better performance with fewer parameters than current models (ViViT-L, Tformer-L and so on) for each dataset (Kinetics 400/600 and Something-Something V2)

Overall, I believe the paper makes a strong contribution and should be of interest to the wider community.

Questions/concerns:

- I wonder how much the training time is different due to the prototype selection runtime bottleneck. How big is the time difference to get results in table3 (c) for 16, 64, and 128 respectively?
- Although it's already explained in the paper, I want to know how slow the training speed of the proposed method is compared to other models. I wonder how long it took to train each baseline model and the proposed model to get the last table result.
- I am curious how the proposed trajectory attention mechanism can be applied to other tasks (e.g., human trajectory prediction). Some commentary on the generality of the proposed modeling contributions on other dynamic tasks would be useful.



**Time Spent Reviewing:**

2 hours

---

> ### Author Response · Authors · 2021-08-10
> **Response to Reviewer m5y4**
>
> Thank you for your review, and your characterization of our work as making a “strong contribution” that “should be of interest to the wider community”. We will consider each of your questions or concerns below.
>
> **Training time**: Overall, our model trains as fast as comparable methods in the literature, while attaining better performance.
>
> **Training time (prototypes)**: For Table 3c, 16 prototypes take 384 GPU hours, 64 prototypes take 800 GPU hours, and 128 prototypes take 1216 GPU hours. We will add these numbers to Table (3c) in the revised paper.
>
> **Training time (baselines)**: For the Kinetics-400 table, the Mformer-B model took 384 GPU hours, the Mformer-L took 1334 GPU hours, and  Mformer-HR model took 1376 GPU hours to train. Our baseline Mformer-B model, which outperforms TimeSformer-B [1] by over 1%, takes similar GPU hours (416 (ours) vs the 416 GPU hours reported in Table 2 of TimeSformer). We cannot directly compare to ViViT [2] because they didn't report training time, but they used a very large transformer (24 layers) compared to ours (12 layers) and so we expect the training time for their approach to be significantly greater.
>
> [1] Is Space-Time Attention All You Need for Video Understanding? Gedas et al. ICML 2021.
>
> [2] ViViT: A Video Vision Transformer. Arnab et al. ArXiv 2021.
>
> **Applications of trajectory attention**: We expect the proposed trajectory attention mechanism to be useful for many tasks involving video data since it has a more favorable inductive bias than existing video attention mechanisms, particularly for dynamic scenes captured by a moving camera. There is significant potential for using trajectory attention for tracking, trajectory prediction, temporal action localization and online action detection, among other settings. For these tasks, the temporal information can be even more important than for action recognition, and so trajectory attention is likely to be especially suitable. We will add a commentary on these applications in the revision.

---

### Decision · Program_Chairs · 2021-09-27

**Decision:**

Accept (Oral)

**Comment:**

This paper presents work on attention in video transformers.  In particular, a novel trajectory-focused self-attention approach is developed, which essentially tracks space-time patches.  The reviewers appreciated the novel method, clear presentation, and effectiveness in empirical evaluations.  The new approach forms a drop-in block that could be used in video architectures.  The reviewers were unanimous in recommending acceptance for the paper.